# Cannabinoids in Late Life Parkinson’s Disease and Dementia: Biological Pathways and Clinical Challenges

**DOI:** 10.3390/brainsci12121596

**Published:** 2022-11-22

**Authors:** Alana C. Costa, Helena P. G. Joaquim, João F. C. Pedrazzi, Andreia de O. Pain, Gustavo Duque, Ivan Aprahamian

**Affiliations:** 1Laboratory of Neuroscience (LIM-27), Departamento e Instituto de Psiquiatria, Hospital das Clínicas HCFMUSP, Faculdade de Medicina da Universidade de São Paulo, São Paulo 05403-903, Brazil; 2Instituto Nacional de Biomarcadores em Neuropsiquiatria (INBioN), Conselho Nacional de Desenvolvimento Científico e Tecnológico, São Paulo 05403-010, Brazil; 3Department of Psychiatry, Faculdade de Medicina da Universidade de São Paulo, São Paulo 01246-903, Brazil; 4Department of Neurosciences and Behavioral Sciences, School of Medicine of Ribeirão Preto, University of São Paulo, São Paulo 05403-903, Brazil; 5Group of Investigation on Multimorbidity and Mental Health in Aging (GIMMA), Geriatrics Division, Department of Internal Medicine, Jundiaí Medical School, Jundiaí 13202-550, Brazil; 6Division of Geriatric Medicine, Research Institute of the McGill University Health Centre, Montreal, QC H4A 3J1, Canada; 7Department of Psychiatry, University Medical Center Groningen, University of Groningen, 9712 Groningen, The Netherlands

**Keywords:** cannabis, cannabinoids, THC, CBD, neurological disorders, psychiatric disorders, elderly

## Abstract

The use of cannabinoids as therapeutic drugs has increased among aging populations recently. Age-related changes in the endogenous cannabinoid system could influence the effects of therapies that target the cannabinoid system. At the preclinical level, cannabidiol (CBD) induces anti-amyloidogenic, antioxidative, anti-apoptotic, anti-inflammatory, and neuroprotective effects. These findings suggest a potential therapeutic role of cannabinoids to neurodegenerative disorders such as Parkinson’s disease (PD) and Alzheimer. Emerging evidence suggests that CBD and tetrahydrocannabinol have neuroprotective therapeutic-like effects on dementias. In clinical practice, cannabinoids are being used off-label to relieve symptoms of PD and AD. In fact, patients are using cannabis compounds for the treatment of tremor, non-motor symptoms, anxiety, and sleep assistance in PD, and managing responsive behaviors of dementia such as agitation. However, strong evidence from clinical trials is scarce for most indications. Some clinicians consider cannabinoids an alternative for older adults bearing Parkinson’s disease and Alzheimer’s dementia with a poor response to first-line treatments. In our concept and experience, cannabinoids should never be considered a first-line treatment but could be regarded as an adjuvant therapy in specific situations commonly seen in clinical practice. To mitigate the risk of adverse events, the traditional dogma of geriatric medicine, starting with a low dose and proceeding with a slow titration regime, should also be employed with cannabinoids. In this review, we aimed to address preclinical evidence of cannabinoids in neurodegenerative disorders such as PD and AD and discuss potential off-label use of cannabinoids in clinical practice of these disorders.

## 1. Introduction

Symptomatic management through commonly prescribed medications and the pharmacological treatment of prevalent disorders can be challenging due to undesirable adverse effects in older adults. The former can be exemplified by the use of opioids for chronic pain, short-term benzodiazepines for anxiety, and antipsychotics for mood disorders or agitation in dementia. Opioid use can result in constipation, confusion, falls, and fractures [1]. Benzodiazepines also raise the risk of falls and fractures due to sedation [2]. Antipsychotics facilitate the development of metabolic syndrome, dyskinesis, and parkinsonism and can contribute to a higher risk of stroke and mortality in people with dementia [3]. The latter condition, pharmacological management of common disorders, can be illustrated by the use of antidepressant drugs for major depression in frail older adults. Antidepressants increase the risk of falls in these patients [4]. To this end, there are clinical and research interests for alternative treatments which could be safer for older adults, especially those with frailty, multimorbidity, or polypharmacy.

The use of cannabinoids as therapeutic drugs has increased among aging populations recently [5], yet, only limited data are available on any age-related differences in cannabinoid effects and clinical research concerning older adults is still scarce [6,7]. Age-related changes in the endogenous cannabinoid system could influence the effects of therapies that target the cannabinoid system [6]. Aging generally seems to result in decreased availability of cannabinoid receptor 1 (CB1) binding sites depending on the brain region [8]. The present narrative review aims to revise the potential mechanisms and biological pathways involving the cannabinoid system and to analyze the current evidence of cannabinoid use for common old-age neuropsychiatric disorders.

## 2. Cannabinoids Use in Late Life

A better understanding of age-related changes in CB1 receptor expression and function and the subsequent changes in behavioral effects of cannabinoid agonists may impact the use of cannabinoids in aging populations. There is increasing interest in the therapeutic use of cannabinoids such as cannabidiol (CBD), synthetic tetrahydrocannabinol (THC) and Cannabis extract, among the aged for various indications including pain, inflammation and multiple sclerosis [6,9,10].

Research about cannabis compounds use among older adults is increasing. Health conditions commonly researched concerning cannabis use among older adults include pain management [11], sleep assistance [12], appetite stimulation [13], and managing behaviors of dementia such as agitation [14]. Data from a survey of 568 volunteers (>years) showed that, for the ones who started using cannabis later in life, it was closely connected to medicinal purpose for issues such as pain management, sleep improvement, and to address anxiety and depression symptoms [15]. Interestingly, cannabis has been employed to replace both prescribed or over-the-counter medications [16,17].

Those data corroborate with research that explored beliefs toward cannabis use. Sixty percent of the older adults surveyed, strongly agreed that the use of medical cannabis was acceptable [18], but he favorability of cannabis decreased as age increased. Notwithstanding, most of older adults consider recreational cannabis as risky and a potential gateway drug [18]. In addition, another study showed that older adults who use cannabis medically or recreationally recognize that there is still a stigma attached to cannabis use regardless of its legality [19]. On the other hand, older adults are less worried about the potential perceived risk of using cannabinoids. Between 2015 and 2019, older Americans showed an 18.8% relative decrease in the perceived risk [20]. 

### 2.1. Cannabinoid Systems and the Brain

The endocannabinoid system (ECS) is the most widespread endogenous signaling neurotransmitter system in the brain [21,22,23,24,25,26,27]. This system can regulate feeding behavior, memory, anxiety, and stress response [28,29,30].

The discovery of the ECS is relatively recent. From experiments carried out with molecules isolated from the plant, it was observed that delta-9-tetrahydrocannabinol (Δ9-THC), through its connection with CB1 receptors, is responsible for the neuropsychological and psychopathological effects [22]. These findings triggered countless other studies that allowed the cloning of the cannabinoid receptor 2 (CB2) receptor [23] and the identification of endogenous molecules that compose the system [24]. The ECS consists mainly of cannabinoid receptors CB1 and CB2; endogenous ligands anandamide (AEA) and 2-arachidonoylglycerol (2-AG); synthesis enzymes such as N-acyl phosphatidylethanolamine (NAPE) and diacylglycerol lipases (DAGL) and degradation or reuptake enzymes as fatty acid amide hydrolase (FAAH) and monoacylglycerol lipase (MAGL) [31,32,33,34].

AEA and 2-AG are both endocannabinoids synthesized on demand from arachidonic acid [26]. Once released in the extracellular space, endocannabinoids act near the synthesis as retrograde synaptic messengers at presynaptic receptors [21]. AEA acts as a partial agonist at CB1 and CB2 [35], but also works on selective cation channels. Transient receptor potential cation channel subfamily V member (TRPV1), a key element in inflammatory conditions and pain [36]. AEA has a notable role in several physiological and neurobehavioral processes, such as pain perception [37], emotional behavior [38] and energy metabolism [39]. 2-AG is the most abundant endocannabinoid in the brain and is considered a full agonist of CB1 and CB2 [40]. It has been implicated in numerous physiological processes [41], including several forms of neuroplasticity [42] and its generation and degradation is part of the lipid metabolism [21].

AEA and 2-AG are the most studied and investigated endogenous ligands. These compounds, unlike classical neurotransmitters, are not synthesized at presynaptic terminals or stored in vesicles but are formed based on demand at postsynaptic terminals. AEA and 2-AG act on presynaptic CB1 or CB2 receptors, inhibiting neurotransmitter release. Because the ECS is widely present in the central nervous system, it plays an essential role in the neurobiology of neurodegenerative diseases such as Alzheimer’s disease (AD) and Parkinson’s disease (PD). Several approaches, whether in vitro assays, animal models, and clinical studies, suggest that ECS modulation can reduce proteins involved in AD pathophysiology, such as tau and beta-amyloid [43] and alpha-synuclein form Lewy bodies in PD. The increased reactivity of microglia and astrocytes, as well as the pro-inflammatory [44] markers TNF-α, IL-17, IFN-γ, iNOS, IL-1β, and NF-κB, are factors implicated in these diseases, where ECS modulation can be a critical pharmacological and molecular target. Furthermore, endocannabinoid modulation can prevent mitochondrial damage, facilitate homeostasis, and decrease excitotoxicity, as well as reactive oxygen species (ROS), culminating in restoring memory and cognitive function, prevalent in the diseases mentioned earlier [45,46,47]. As seen in the image (Figure 1), adequate functioning of the ECS can be an essential tool in the homeostasis of inflammatory responses, in glial reactivity, in the proper functioning of mitochondrial complexes, and the control of the expression of proteins implicated in the pathophysiology of AD (Table 1) and PD. Furthermore, this system and its complex machinery have also participated in synaptic plasticity and neurogenesis events [46]. Both CBD and THC have potential targets for therapeutic effects on neurodegenerative diseases, since they can modulate ECS. CBD acts as an agonist of the receptors TRPV1, PPARγ, and mAChR and as an antagonist of the receptor GPR55 [48,49]. This compound is suggested to act as a negative allosteric modulator of CB1 and CB2 receptors [50]. Finally, CBD inhibits the enzyme FAAH, with a consequent increase in AEA levels. Moreover, AEA can activate CB1, CB2, and TRPV1 receptors (Table 2). CBD is relevant for treating neurodegenerative diseases since it can increase the activity of mitochondrial complexes and shows antioxidant and anti-inflammatory effects that are partially mediated by its actions on TRPV1, mitochondria, and PPARγ. On the other hand, THC is a partial agonist of CB1 and CB2 receptors, an agonist of GPR55 and PPARγ, which, just like CBD, can exert anti-inflammatory effects. Figure 1 summarizes the effects of cannabinoids in dementias.

### 2.2. Translational Research

#### 2.2.1. Pre-Clinical Findings

Small, limited trials have directly examined the effects of cannabinoid-based therapeutics on patients with dementia originating due to various underlying pathologies [46]. Most of the studies that we have available in the scientific literature on the action of cannabinoids on pathological mechanisms of dementia were observed in preclinical models—in vitro and animal.

The pre-clinical experiments show that CB2 activation induces a neuroprotective effect in animal models of vascular dementia. More specifically, CB2 agonism reduces memory impairment and infarct size during cerebral hypoperfusion and vascular dementia [51]. At the preclinical level cannabidiol (CBD) induces anti-amyloidogenic [52], antioxidative, anti-apoptotic, anti-inflammatory [53], and neuroprotective effects [54]. Apart from this, a reduction CB1 receptor binding in CNS has already been described in aged-related populations [8].

#### 2.2.2. In Vitro and In Vivo AD-Related Assays

In PC12 neuronal cells, a valuable strain for the study of nerve cell differentiation, the activation of PPARγ receptors by CBD through the Wnt/β-catenin pathway was able to protect this cell group from Aβ neurotoxicity and oxidative stress, increase survival and reduce ROS production in addition to decreasing lipid peroxidation, and inhibit tau protein hyperphosphorylation and stimulate neurogenesis in the hippocampus [55]. These results suggest that CBD modulation of the endocannabinoid system could be a viable tool to treat or even prevent the course of AD. It is noteworthy that both CBD and Δ9-THC have an affinity for the peroxisome proliferator-activated receptor (PPARγ) and can trigger anti-inflammatory effects and play a role in neuroprotection. Additionally, CBD may be beneficial in treating inflammatory processes associated with Aβ deposition. In mice injected with strains of this cell type and microglia, this drug inhibited the generation of nitric oxide (NO), increased intracellular calcium induced by ATP in cultured microglia, and promoted the migration of microglial cells. Furthermore, CBD prevented Aβ-induced learning deficit and mRNA expression for IL-6 [56]. In addition, in in vitro cultured astrocytes, CBD-induced reduction in FAAH activity was accompanied by increased glycogen phosphorylated kinase 3β (GSK3β) synthesis. Furthermore, it promoted dendritic remodeling and increased hippocampal synaptic protein expression [57]. In vitro-cultured Human Gingival Mesenchymal Stem Cells, the CBD prevented the expression of proteins potentially involved in tau phosphorylation and Aβ production [58]. Interestingly, CBD acted selectively on the activation of PPARγ receptors and reduced the expression of the APP+ protein and its ubiquitination, leading to a reduction of Aβ and neuronal apoptosis in SH-SY5Y APP+ cells [59]. CBD was also able to dose-dependently reduce the effects of Aβ mediated by NF-κB inhibition in rat primary astrocytes. In this same context, in in vivo assays, CBD could also reduce gliosis and glial fibrillary acidic protein [60]. In another in vivo approach conducted on mouse hippocampal slices, CBD improved synaptic transmission and long-term potentiation [61]. In the right dorsal hippocampus of mice treated with the human Aβ (1–42) peptide, the application of CBD significantly inhibited the expression of GFAP, together with its mRNA and protein, in addition to reducing the expression of other pro-inflammatory markers such as proteins iNOS and IL-1β [60]. Interestingly, studies that used the association of CBD and Δ9-THC observed a significant improvement in the memory of AβPP/PS1 transgenic mice, with greater efficacy than CBD or Δ9-THC when administered alone. In addition, the association between CBD and Δ9-THC reduced the expression of GluR2/3 receptors of aged AβPP/PS1 transgenic mice and increased the level of expression of GABA-A Rα receptors [62]. Corroborating these findings, in an assay that used an N2a variant of amyloid β precursor cells (AβPP), were treated with Δ9-THC, this compound was able to cause a neuroprotective effect in this cell group, with a pronounced decrease in Aβ expression [63]. Additionally, in a recent study conducted with Tg4-42 transgenic mice that express Aβ4-42 of human origin, when treated with Δ9-THC, there was a reduction in neuronal loss compared to the control group [64].

#### 2.2.3. In Vitro and In Vivo PD Related Assays

Animal models of PD, both in vitro and in vivo, generally employ the use of neurotoxins such as 6-Hydroxydopamine (6-OHDA) and 1-methyl-4-phenyl-1,2,3,6-tetrahydropyridine (MPTP), which promote dopaminergic neuronal loss in the substantia nigra, similar to what occurs in patients with the disease, in addition, these drugs promote an increase in pro-inflammatory markers, oxidative stress, and excitotoxicity. In an MPTP-induced PD model in Δ9-THC-treated marmoset monkeys, restoration of the locomotor activities to nearly pre-disease level was observed [65]. The assumption is that this positive effect is due to the elevated expression of CB1 receptors in the marmoset PD models. This theory can be supported and validated with the data from a previous study using PD marmoset models with MPTP, where the animals showed higher concentrations of CB1R expressed in the basal ganglia [66]. Although few studies have used CBD in vivo PD models, the findings are promising. In this sense, in a model of PD induced by 6-OHDA in mice, it was observed that Cannabidiol Quinone Derivative VCE-004.8 had a neuroprotective effect on the lesion, increased the expression of tyrosine hydroxylase (TH) and decreased inflammatory markers in the substantia nigra. Additionally, in in vitro assays, the cytoprotective effects of CBD were mediated primarily via PPAR-γ receptors [67]. In a recent study that MPTP was administered to mice. The oral administration of CBD improved behavior in cognitive tasks and spontaneous locomotion, in addition to increasing serotonin and dopamine levels in the substantia nigra. Concurrent with these findings, CBD promoted a decrease in inflammatory markers such as TNF-α, IL-1β, and IL-6. In addition, the animals treated with CBD showed increased tyrosine hydroxylase (TH) expression in the substantia nigra. In addition, they upregulated the expression of Bcl-2, decreasing the levels of Bax and Caspase-3 and preventing the expression of the NLRP3/caspase-1/IL-1β inflammasome pathway [68]. Patients affected with PD, in general, make use of L-3,4-dihydroxyphenylalanine (L-DOPA), a dopamine precursor drug, which helps treat motor impairments at least in the first and middle phases of the disorder, but without interrupting the course of the disease [69]. On the other hand, prolonged use of L-DOPA can cause tardive dyskinesia, a highly disabling effect for patients with the disease [70]. Interestingly, in a preclinical PD model, it was observed that 6-OHDA-lesioned mice with levodopa-induced dyskinesia were simultaneously treated with capsazepine (TRPV1 receptor antagonist) and CBD showed a pronounced decrease in dyskinetic movements. In addition, this association promoted a reduction in pro-inflammatory markers, such as COX-2 expression, in motor regions of the striatum [71]. Similarly, the co-administration of compounds such as HU-210 and nabilone reduced this adverse effect [72,73]. Neurodegenerative diseases such as AD and PD have a robust involvement with mechanisms related to oxidative stress. In general, compounds that modulate the endocannabinoid system have antioxidant properties, making them viable tools for investigating these effects and underlying mechanisms in these diseases. In this sense, compounds such as CBD and AM404 were able to reduce these effects in 6-OHDA-induced nigrostriatal lesions in rats [74]. Similarly, the CB2 receptor agonist HU-308 had a similar effect on this type of injury. Moreover, the action of compounds via CB1, such as WIN55, 212-2, and the CB2 receptor agonist JWH015 were able to reduce microglial activation in MPTP-induced injury in mice L-DOPA-induced dyskinesia [75]. So far, the results of preclinical trials using cannabinoids, especially CBD, in models of neurodegenerative diseases such as PD and AD are encouraging. However, there is a need to expand investigations to elucidate these drugs’ pharmacological and molecular mechanisms in preventing, course, and mitigating the damage caused by these diseases.

### 2.3. Clinical Findings

#### 2.3.1. Parkinson’s Disease (PD)

Previous studies were published that directly examined endocannabinoid system-induced effects on PD patients. The underlying pathology of PD typically involves complex clinical manifestations such as oxidative stress, neuro-inflammation, and pain [76]. As such, any novel treatment strategies ought to be able to address this complexity by simultaneously targeting multiple pathways and mechanisms to stop and/or slow down the neurodegenerative processes. Cannabinoids acting on classical cannabinoid receptor sites (CB1 and CB2) and a variety of other cannabinoid-sensitive receptor sites (e.g., TRPV1, Peroxisome proliferator-activated receptor gamma—PPARγ, dopamine, glutamate, GABA) have demonstrated a considerable therapeutic impact by attenuating signs and symptoms such as inflammation, oxidative stress, pain, stress, movement disorders—tremors, rigidity, bradykinesia, L-DOPA-induced dyskinesia [77], mood disorders (e.g., depression), and insomnia [78]. Adverse effects such as changes in cognition, ataxia, motor skills, dysphoria, and dependence are typically THC-dose-dependent [79,80].

Two recent systematic reviews and meta-analyses evaluated the impact of cannabis compounds on movement disorders and Parkinson’s disease [81,82]. Few studies were double-blind randomized controlled studies. Current evidence does not support a high level of recommendation for the use of cannabis. Two clinical trials investigated cannabis in motor symptoms but found no improvement [82]. However, several potential benefits were observed in decreasing tremor, anxiety, and pain. Additionally, improvement in sleep quality and quality of life was also observed. In particular, trials with nabilone, a THC analog, improved quality of life, levodopa-induced dyskinesia, anxiety, anxiety-induced tremors, and sleeping problems. CBD use also showed promising results, although involving small samples with short follow-up, and almost all evidence come from the same research group from Brazil. An open-label study with six PD patients with psychotic symptoms using antiparkinsonian medications tested CBD doses ranging from 150–400 mg/day [83]. CBD was safe and reduced psychotic symptoms according to different scales (the Brief Psychiatric Rating Scale and the Parkinson Psychosis Questionnaire). No influence on cognitive and motor signs was found. The same group published a case series of four PD patients with REM sleep behavior disorder [84]. All patients showed rapid and substantial reductions of sleep disorder symptomatology after CBD treatment, and this effect disappeared after CBD discontinuation. Finally, this group published an exploratory double-blind trial of CBD versus placebo involving twenty-one PD patients without dementia [85]. Patients were assigned to three groups of seven participants: placebo, CBD 75 mg/day, or CBD 300 mg/day. Authors aimed to evaluate motor and general symptoms (Unified Parkinson’s Disease Rating Scale [UPDRS]) and well-being and quality of life (Parkinson’s Disease Questionnaire [PDQ-39]). Although no differences were observed across groups in motor scores, participants treated with CBD 300 mg/day had significantly different scores in the PDQ-39, reflecting a potential role of CBD for improving quality of life of PD patients.

#### 2.3.2. Alzheimer’s Disease (AD) and Other Dementias

Data suggest that cannabinoid-based therapeutics acting on cannabinoid-sensitive receptor sites [e.g., CB1, CB2, PPARγ, TRPV1, mAChR (muscarinic acetylcholine receptor)] or by modulating endocannabinoids (e.g., AEA, 2-AG and/or their respective metabolizing enzymes FAAH and MAGL) may produce AD-relevant therapeutic effects by modulating abnormal processing of Aβ and tau, by reducing neuroinflammation, excitotoxicity, oxidative stress, and mitochondrial dysfunction, by protecting from neuroinflammation-induced cognitive damage, and by restoring memory and cognitive function (in test animals) [86], while simultaneously supporting the brain’s intrinsic repair mechanisms by augmenting neurotrophin expression and enhancing neurogenesis [55,87]. No trials reported that cannabinoid-based therapeutics worsen AD pathologies.

A recent Cochrane systematic review was published to identify any potential benefits of cannabis use in dementia [88]. Primary outcomes in this systematic review were changes in cognitive function, behavioral and psychological symptoms of dementia (BPSD), and adverse events. The review was restricted to only four clinical trials involving 126 participants, mostly diagnosed with AD followed by vascular and mixed (AD and vascular) dementia. Most studies showed a low risk of bias. All studies used THC-based therapies, the natural Δ9-THC and two synthetics, dronabinol and nabilone, between 3 to 14 weeks. Adverse events were reported, including one trial with more than 70 weeks of follow-up. The authors reported insufficient evidence of cognitive improvement with THC, translating into a potential mean reduction of 1 point in the standardized Mini-Mental State Examination [89]. A similarly low level of evidence was found concerning the removal of BPSD. They observed a mean decrease of fewer than 2 points in the Neuropsychiatric Inventory. Adverse events were not reported to permit meta-analysis, and the authors judged those reports to be imprecise. In general, THC-based therapy was safe, without severe adverse effects [90]. Only sedation and lethargy were almost three times more common in participants receiving nabilone.

In addition, cannabis research in dementia has been directed towards a major interest in managing cognitive decline and BPSD; one previous trial evaluated the severe loss of appetite that some patients present even during the initial stages of AD. A small sample study with probable AD demonstrated that Dronabinol use resulted in weight gain and decreased BPSD compared to placebo [91].

#### 2.3.3. Management Challenges

Currently, it is not unusual that patients and/or clinicians consider cannabinoids as an alternative for older adults with neuropsychiatric diseases and a poor response to first-line treatments [11,92,93,94]. Several countries such as the United States (USA), for example, permit the prescription of cannabinoids from a licensed healthcare provider and have approved cannabinoids for therapeutic use (e.g., USA Food and Drug Administration approved the CBD compound Epidiolex for Lennox–Gastaut syndrome or Dravet syndrome). In our concept and experience, cannabinoids should never be considered as a first-line treatment but could be applied to specific situations commonly seen in clinical practice or as an adjuvant therapy with first-line or second-line treatments for neuropsychiatric disorders of late-life such as AD or PD. The main reason sustaining this argument is the lack of high-quality evidence from clinical trials involving cannabinoids in several late-life neuropsychiatric disorders. Whereas cannabis evidence lacks a compelling strong recommendation for PD, AD, and other dementias, we share practical therapeutic orientations of when and how to consider prescribing cannabinoids in these situations (Table 2). We have to be aware that cannabinoids have not been properly tested for their pharmacological properties (pharmacokinetics and pharmacodynamics) in older adults. So, questions regarding safety, efficacy, biodisponibility, duration of use are depicted from small clinical trials, cohort, and cross-sectional studies. This is a major issue when considering cannabinoids for older adults. However, prescription of cannabinoids is increasing among older adults. Unfortunately, no previous article has discussed practical clinical utility of these compounds. These disorders, such AD and PD, result in a high burden for patients, family members, and caregivers. They often are associated with resistant symptomatology after first- or second-line treatment recommendations are implemented. Finally, evidence from clinical trials frequently over-select these patients, which is not always a reflection of real-world cases that demand alternative treatments for the improvement of symptoms or quality of life of the patient and family. In this scenario, cannabinoids are potential candidates due to their positive and emerging pre-clinical evidence and their favorable safety profile compared to psychotropic medications, for example.

Unfortunately, many prescribers do not have access to reliable recommendations from experts or have access to good peer-reviewed publications. Mostly prescribers base their prescription on the dispenser’s suggestions. A systematic review of cannabidiol in clinical populations was published two years ago [95]. Twenty-three studies with a low risk of bias were included, reporting doses varying between <1 to 50 mg/kg/daily. The studies only included cannabidiol extracts or oils, and a mean dose of 2.4 mg/kg/daily could be yielded from these studies. Another systematic review evaluated the efficacy and safety of cannabinoids specifically in older people [96]. Five cross-over clinical trials with 267 participants were included, but one reported a mixed population of younger adults. Three studies used a THC formulation, 5 mg to 60.5 mg, and two used a mix of THC and cannabidiol. Two studies showed that THC was helpful in the treatment of anorexia and BPSD. There are no large studies evaluating pharmacokinetic of pharmacodynamic properties of CDB, THC or synthetic THCs in older adults. Older adults have physiological changes that directly impact pharmacological principles such as half-life, liver metabolism, and renal clearance. Since cannabinoids are not without adverse effects and can produce side reactions to their prescription, pharmacological studies are essential in the elderly population.

Currently, no guidelines are published concerning the prescription of cannabinoids for older people. In 2021, a consensus recommendation on dosing and administration of cannabis compounds for chronic pain was published [97]. This publication involved several international experts representing nine countries. Using a modified Delphi process, three treatment protocols were proposed, including one specifically dedicated to vulnerable patients with higher risk of adverse reactions. Essentially, protocols propose initiating with pure CBD at 5 mg with progressive increasing until 40 mg daily. THC was recommended to be added after 20 mg of CBD, and a progressive weekly increase was also recommended. The conservative protocol, aimed to vulnerable people, which we identify has been ideal to older adults with neuropsychiatric disorders, recommends a 5 mg once daily CBD-predominant compound increased by 10 mg every 2 to 3 days up to 40 mg/day. At 40 mg/day, adding THC at 1 mg/day and titrate by 1 mg every 7 days until 40 mg/day is an option to achieve the goals. Although these protocols are not intended to older adults with neuropsychiatric disorders, we consider it as a potential starting point for prescribing those patients due to its low starting dose and progressive tapering to achieve therapeutic doses. 

The physiological changes associated with aging (e.g., decreased organ function, impaired cognitive function, decreased fat-free body mass) may increase the risk or magnitude of adverse and impairing effects related to cannabis consumption.

Typically, there is a greater risk for drug interactions in this population [98]. Thus, this population generally requires more frequent monitoring. A total of 36.7% of adverse events were reported among cannabinoid users compared to placebo, according to a systematic review of older adults [99]. To mitigate the risk of adverse events, the traditional dogma of geriatric medicine, starting with a low dose and a slow titration regime, should also be employed with cannabinoids. THC results in more frequent adverse reactions than CBD compounds, most probably due to psychoactive properties [99]. Sedation, asthenia, nausea, vertigo, drowsiness, fatigue, euphoria, anxiety, and emotional lability were reported using cannabinoids. A prospective study with real-world evidence in a specialized cannabis clinic observed lower percentages of adverse reactions in older adults [11]. A total of 2736 patients with a mean of 74 years were evaluated in 2 years of follow-up. Of this sample, 182 had neuropsychiatric disorders. Five adverse symptoms occurred more than 2%: dizziness (9.7%), dry mouth (7.1%), somnolence (3.9%), weakness (2.3%) and nausea (2.2%). 

Finally, although the safety profile is potentially better than several common drugs older adults use, attention must be paid to potential drug interaction mainly involving the cytochrome P450 system [100,101]. The use of anti-inflammatories, anticonvulsants, asthma medications (e.g., zafirlukast), chemotherapies (e.g., cyclophosphamide), hypoglycemic drugs, anticoagulants (e.g., warfarin), and ACE inhibitors combined with cannabinoids demands further vigilance by the prescriber. 

## 3. Conclusions

Cannabinoids constitute a promising pharmacological approach to treatment of neuropsychiatric disorders in late life. However, evidence from high-level clinical trials is lacking and these compounds should never be used as first or second-line therapies. Their use should be restricted to adjuvant off-label treatment after approved, high-level evidence recommendations are implemented. Moreover, pharmacological studies, especially dedicated to efficacy and safety of cannabinoids in older people are urgently needed. Controlled trials with longitudinal designs and larger samples are required to examine the long-term efficacy of these drugs in dementia, AD and PD. Overall, cannabinoids compounds are well tolerated and appear to be safer than most psychotropic medication, but given the vulnerability of patients with dementia, they require appropriate monitoring by the clinician. At an off-label prescription level of cannabinoids to older people, a geriatric principle of prescription of starting low doses with slower titration should be done.

## Figures and Tables

**Figure 1 brainsci-12-01596-f001:**
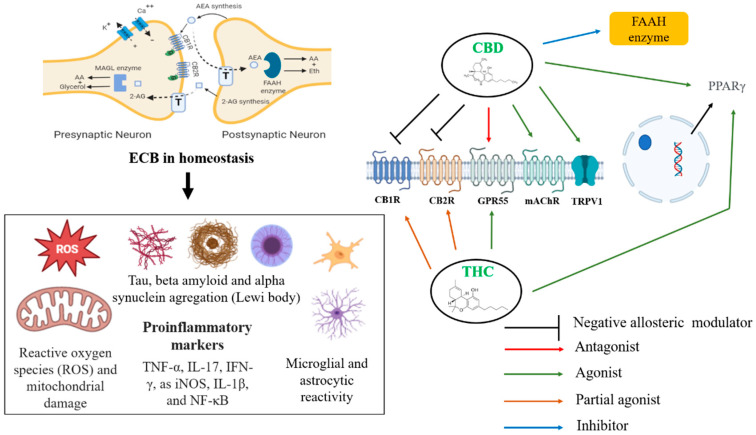
Potential targets and therapeutic effect of CBD in dementias. Legend: TRPV1, transient receptor potential vanilloid type 1; PPARγ, peroxisome proliferator-activated receptor gamma; GPR55, G-protein-coupled receptor 55; CB1, cannabinoid receptor type 1; CB2, cannabinoid receptor type 2; FAAH, fatty acid amide hydrolase; mAChR, muscarinic acetylcholine receptor; 2-AG, 2-arachidonoylglycerol; AEA, anandamide; T, transporter.

**Table 1 brainsci-12-01596-t001:** Potential targets and therapeutic effect of CBD in dementias.

Receptors	Action	Pharmacology Propriety
CB1	Direct antagonist and negative allosteric modulation antagonist	Attenuation of learning deficit, memory, and psychotic effects of THC
CB2	Antagonist & reverse agonist	Anti-inflammatory
GPR55	Antagonist	Antitumor
5HT1A	Agonist	Analgesia and anxiolytic
mAChR	Agonist	Cognition improvement
TRPV1	Agonist	Anti-inflammatory and analgesia
PPARγ	Agonist	Antioxidant and anti-inflammatory

Legend: CB1 Cannabinoid receptor type 1; CB2, cannabinoid receptor type 2; GPR55, G-protein-coupled receptor 55; 5HT1A Serotonin 1A receptor; mAChR, muscarinic acetylcholine receptor; TRPV1, transient receptor potential vanilloid type 1; PPARγ, peroxisome proliferator-activated receptor gamma.

**Table 2 brainsci-12-01596-t002:** Practical management with cannabinoids in Parkinson’s and Alzheimer’s disease.

Neuropsychiatric Disorder	Potential (Off Label) Indication	Suggested Dose Regimen
Parkinson’s disease	Resistant tremor or dyskinesia	Starting dose: CBD (<0.3% THC) 5 mg once daily. Increase 5 mg every 3 days. Maximum dose: 20 mg twice a week.
	Resistant anxiety	Starting dose: CBD (<0.3% THC) 5 mg once daily. Increase 5 mg every 3 days. May split the dose in two or three intakes. Maximum dose: 90 mg twice a week (CBD monotherapy). 1 mg of THC can be initiated with CBD or after 20 mg of CBD without a positive effect. Increase THC to a maximum of 20 mg combined to a maximum of 40 mg of CBD.
	Agitation due psychosis partially treated with quetiapine or clozapine	
	Persisted sleeping disturbance albeit treated with two first-line treatment	Starting dose: CBD (<0.3% THC) 5 mg at night. Increase 5 mg every 3 days. Maximum dose: 20 mg
Alzheimer’s disease	Persisting agitation or aggression besides non-pharmacologic and first-line drug treatment implemented	Starting dose: CBD (<0.3% THC) 5 mg once daily. Increase 5 mg every 3 days. May split the dose in two or three intakes. Maximum dose: 20 mg twice a week. 1 mg of THC can be initiated with CBD or after 20 mg of CBD without a positive effect. Increase THC to a maximum of 20 mg
	Major adverse event with first-line drug treatment for agitation, anxiety, or aggression	
	Persisting anorexia albeit traditional treatment for dementia and exclusion of secondary causes	Starting dose: CBD (<0.3% THC) 5 mg once daily. Increase 5 mg every 3 days. Maximum dose: 10 mg twice daily

Note: CBD = cannabidiol; THC = delta-9-tetrahydrocannabinol.

## Data Availability

Not applicable.

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
