# Peer review of "Cannabinoids in Late Life Parkinson’s Disease and Dementia: Biological Pathways and Clinical Challenges"

_brainsci, 2022, doi:10.3390/brainsci12121596_

Round 1

Reviewer 1 Report

find in attachment

Author Response

Dear Reviewer,

We have uploaded our reply. Many thanks for your time.

Regards

Reviewer 2 Report

The article «Cannabinoids in late life neuropsychiatric disorders: biological 2 pathways and clinical challenges» is of undoubted interest to clinicians, contains a broad overview of the data presented in a logical sequence.

Author Response

(The authors gave the same response as above.)

Reviewer 3 Report

The efficacy of cannabinoids in the treatment of childhood epilepsy syndromes is increasingly recognized. The possibility has also been discussed in recent years, the medical cannabinoids to be used in the treatment of older people. There are concerns about this, but they have the potential role to be helpful (but not first-line treatment) in  some conditions.

The risks of cannabinoids in older patients appear to be moderate and their incidence is comparable to other classes of analgesic drugs. However, the quality and quantity of the studies is low, and few older patients have been included in cannabinoid studies. Special studies are needed to determine the effectiveness and safety of cannabinoids in older patients.

The study covers a good number references, but there are others that could be added. Those that are considered by the authors, on what principle are they selected? I think there is a sufficient amount of information that would help the authors to expand their horizons and supplement the data in the article.

Here are some other comments, that authors need to go through them:

1.      The abstract not clearly illustrated the aims of this article. But if it can provide more specific words, it would prompt readers to take a few minutes more to reconsider whether or not he/she should spend time reading the whole article. As for the title, it covers quite a large area of neuropsychiatric diseases, but two are mainly discussed (AD and PD). It is informative enough to draw its potential reader's attention, but is not very precise as it is limited to certain diseases. Common forms of dementia associated with neurodegeneration are also Huntington's disease and Vascular dementia. Perhaps some information about them would improve and expand the article.

2.      In the introduction of this article is successfully explained why current research is important. Nevertheless, the aim to revise the potential mechanisms and biological pathways involving the cannabinoid system and to analyze the current evidence of cannabinoid use for common old-age neuropsychiatric disorders (line 59-61). The following points should also be considered in the manuscript according to the context (may be in this chapter): to present a diagram/figure about the types of cannabinoids and possible effects. Their action also affects other systems such as e.g. endocannabinoid, opioid, serotonin, acetylcholine, etc. and can have different effects. Some of them change the activity of receptors similar to aging and affect cognitive abilities, but in others they have a balancing effect. It does not necessarily need to be commented on, but these data will clarify the reasons why the choice for an adjuvant therapy with cannabinoids in specific situations in clinical practice is possible.

3.      In this part it is necessary to include  more appropriate references to related and previous works. There are only 6 references in the chapter, which is extremely few. All these refinements will make the article more interesting for readers from multiple backgrounds.

4.      Line 58 - …. „Aging generally seems to result in decreased availability of cannabinoid receptor 1 (CB1) ......” – reference?

5.      Chapter 1.2. "Cannabinoid systems and the brain" – Cannabinoid receptors, agonists, locations, and actions could be given in table.

6.      Line 131, 137,142 …. references?

7.      In Pre-clinical findings (line 157). Two small information is given. The authors could amplify this chapter by using more references (example: Walther S, Halpern M. Cannabinoids and Dementia: A Review of Clinical and Preclinical Data. Pharmaceuticals (Basel). 2010 Aug 17;3(8):2689-2708. doi: 10.3390/ph3082689; Liu CS, Chau SA, Ruthirakuhan M, et al. Cannabinoids for the treatment of agitation and aggression in Alzheimer’s Disease. CNS Drugs. 2015;29(8):615–623; Boggs DL, Nguyen JD, Morgenson D, Taffe MA, Ranganathan M. Clinical and Preclinical Evidence for Functional Interactions of Cannabidiol and Δ9-Tetrahydrocannabinol. Neuropsychopharmacology. 2018 Jan;43(1):142-154. doi: 10.1038/npp.2017.209; Bailey, M.M., Emily Mills, M.C., Haas, A.E. et al. The effects of subacute exposure to a water-soluble cannabinol compound in male mice. J Cannabis Res 4, 44 (2022). https://doi.org/10.1186/s42238-022-00153-w; Bailey, M.M., Emily Mills, M.C., Haas, A.E. et al. The effects of subacute exposure to a water-soluble cannabinol compound in male mice. J Cannabis Res 4, 44 (2022). https://doi.org/10.1186/s42238-022-00153-w .........................)

8.      Line 168, 173,179, 182,183…..references

9.      Line 207 – 215, 224 – 226….. – references

10.   In chapter “Management challenges”  - Are there potential indications for medical cannabinoids in the older person? Discus the problems with the availability of standard protocols, doses and possible duration of treatment. The authors can provide some suggestions/solutions/capabilities for them. Complete the data from Table 1 if there is information on other types of dementia (Huntington’s disease and Vascular dementia).

Author Response

(The authors gave the same response as above.)

Round 2

Reviewer 1 Report

The manuscript was improved a lot, the following minor points needs to be clarified before the final decision:

66. I think there is a misinterpretation of the compound names: delta 9-tetrahydrocannabinol, briefly THC is a molecule having absolutely the same effect whether synthesized or extracted from a plant. Nevertheless, cannabis extracts contain besides THC always also other phytocannabinoids in physiologically/pharmacologically relevant concentrations, therefore this extract has obviously different properties than pure THC. Thus, the authors should write CBD, synthetic THC and cannabis extract in this sentence to avoid misunderstanding. If they want to explain that the cannabis extract contains THC it is up to them.

67. The indication of THC-treatment is very seldom memory or cognitive impairment: the study the authors cited is about memory impairment as side effect in patients treated with THC to alleviate symptoms of multiple sclerosis.

106-109 The sentence starting “Those mechanisms” is misplaced here and not connected at all with the previous sentence.

144 Please re-phrase the sentence as follows: … adequate functioning of the ECS is essential in the regulation of inflammatory responses, …

218 Reference is missing to the superior effect of combined THC, CBD treatment

406 This sentence suggest that 40 mg/day is the dose of the THC. Please re-phrase, I understood only from your response that you mean CBD.

Author Response

Thanks for your suggestions. We have replied in the attached file.
